# Photocatalytic Study of Cyanide Oxidation Using Titanium Dioxide (TiO₂)-Activated Carbon Composites in a Continuous Flow Photo-Reactor

**Stalin Coronel, Diana Endara \*, Ana Belén Lozada, Lucía E. Manangón-Perugachi**  **and Ernesto de la Torre**

Department of Extractive Metallurgy, Escuela Politécnica Nacional, Ladrón de Guevara E11-253, Quito 170517, Ecuador; stalin.coronel@epn.edu.ec (S.C.); ana.lozada@epn.edu.ec (A.B.L.); lucia.manangon@epn.edu.ec (L.E.M.-P.); ernesto.delatorre@epn.edu.ec (E.d.l.T.)
\* Correspondence: diana.endara@epn.edu.ec; Tel.: +593-(9)9854-9231

**Abstract:** The photocatalytic oxidation of cyanide by titanium dioxide (TiO₂) supported on activated carbon (AC) was evaluated in a continuous flow UV photo-reactor. The continuous photo-reactor was made of glass and covered with a wood box to isolate the fluid of external conditions. The TiO₂-AC synthesized by the impregnation of TiO₂ on granular AC composites was characterized by inductively coupled plasma optical emission spectrometry (ICP-OES), Scanning Electron Microscopy (SEM), and nitrogen adsorption-desorption isotherms. Photocatalytic and adsorption tests were conducted separately and simultaneously. The results showed that 97% of CN⁻ was degraded within 24 h due to combined photocatalytic oxidation and adsorption. To estimate the contribution of only adsorption, two-stage tests were performed. First, 74% cyanide ion degradation was reached in 24 h under dark conditions. This result was attributed to CN⁻ adsorption and oxidation due to the generation of H₂O₂ on the surface of AC. Then, 99% degradation of cyanide ion was obtained through photocatalysis during 24 h. These results showed that photocatalysis and the continuous photo-reactor's design enhanced the photocatalytic cyanide oxidation performance compared to an agitated batch system. Therefore, the use of TiO₂-AC composites in a continuous flow photo-reactor is a promising process for the photocatalytic degradation of cyanide in aqueous solutions.

**Keywords:** cyanide; activated carbon; titanium dioxide; composites; continuous flow; photocatalysis; adsorption

## 1. Introduction

Cyanide is a highly toxic pollutant, which, even at low concentration, may cause human health and environmental problems [1]. Cyanide is rapidly and extensively absorbed by the human body through the oral inhalation and dermal routes. It prevents the transport of oxygen, which affects the cellular respiration process, leading to suffocation in the worst case and eventually to death [2].

Cyanide is present in industrial wastewaters such as coal gasification, electroplating, plastics, pharmaceuticals, and the mining industry. These wastewaters are discharged in the water bodies causing serious threats to the environment [1,3,4].

In Ecuador, the artisanal and small-scale gold mining activities make significant contributions to mineral production [5,6]. Large-scale gold mining projects as "Fruta del Norte" and "Cascabel" have been developed in the last years. Both small- and large-scale gold mining industries use cyanidation to recover gold from ores.

In Ecuador, cyanidation and carbon in pulp (CIP) processes use an aqueous 500 mg/L NaCN solution. The Ecuadorian legislation TULSMA establishes a discharge limit of 1 mg/L total cyanide into sewers and 0.1 mg/L into surface fresh waters [7]. Cyanides exist in the form of free ions (CN⁻) and also can form complexes, which makes its treatment more difficult. For example, a study with gold mining wastewater that contained copper

and zinc was treated with zeolite, achieving a 93.97% degradation efficiency of total cyanides [8]. Due to the hazard of cyanide, its treatment is very important. Several physical, biological, and chemical treatment processes have been applied to remove cyanide. The most commonly used processes include chemical oxidation, alkaline chlorination, hydrogen peroxide oxidation, INCO process (purification with $SO_2$ and air), oxidation with Caro acid, and ozonation. Chlorination is the most used process, and it is very efficient in the elimination of cyanide; however, it has certain disadvantages, such as the generation of toxic intermediate compounds which must be treated, making the process expensive [9,10].

Advanced oxidation processes (AOPs) have been studied extensively in the removal of different contaminants. Hydroxyl radicals are considered the most reactive oxygenated species within AOPs due to their high oxidation potential and their non-selective nature. Photocatalysis is an effective technique for treating toxic substances, including cyanide. [11,12].

Several semiconductor materials have been tested as photocatalysts for the removal of aqueous pollutants. However, difficulties related to the stability of the photocatalyst under irradiation in water have been evidenced. It is accepted that titanium dioxide $TiO_2$ in anatase phase is the most reliable photocatalyst for aqueous pollutant removal [13].

In heterogeneous photocatalysis based on semiconductors, the photocatalyst $TiO_2$ is excited by absorbing incident UV radiation. Momentarily, the electron of the valence band is transferred to the conduction band, and the electron/hole pair ($e^-/h^+$) is formed. Then, the electron/hole pair reacts with water and dissolved molecular oxygen to generate hydroxyl and superoxide radicals, which are responsible for the photocatalytic oxidation of free cyanide [14].

Direct and indirect mechanisms of photocatalytic oxidation of free cyanide with $TiO_2$ have been proposed. In the direct mechanism, free cyanide present in the solution is oxidized directly through the transfer of electrons from the holes of the valence band. The indirect mechanism occurs through the adsorbed $OH^\bullet$ radicals, implying that the contaminants are first adsorbed on the photocatalyst surface and then react with the excited superficial $e^-/h^+$ pairs or the $OH^\bullet$ radicals [15].

The photocatalytic oxidation of cyanide starts with the formation of cyanide radical, which dimerizes to form cyanogen. Then, the cyanogen undergoes transformation in an alkaline medium to give cyanide and cyanate. Finally, cyanate oxidizes to form nitrite ($NO_2^-$), nitrate ($NO_3^-$), carbonate ($CO_3^{2-}$), carbon dioxide ($CO_2$), and nitrogen ($N_2$). These reactions are shown in the Equations (1)–(5) [16].

$$CN^- \overset{h^+/OH^\bullet}{\rightarrow} CN^\bullet \tag{1}$$

$$2CN^\bullet \rightarrow (CN)_2 \tag{2}$$

$$(CN)_2 + 2OH^- \rightarrow CN^- + CNO^- + H_2O \tag{3}$$

$$CNO^- + 8OH^- + 8h^+ \rightarrow NO_3^- + CO_2 + 4H_2O \tag{4}$$

$$CNO^- + 2H_2O + 3h^+ \rightarrow CO_3^{2-} + \frac{1}{2}N_2 + 4H^+ \tag{5}$$

$TiO_2$ has been widely used as photocatalyst due to its non-toxicity, low cost, chemical stability, and its favorable chemical and physical properties. It can be reused several times without reducing its catalytic activity. $TiO_2$ has been tested in the photocatalytic degradation of wastewater under ultraviolet (UV) light. However, low efficiencies in photodegradation have been achieved due to its rapid unfavorable charge carrier recombination reaction in $TiO_2$ and the high band gap energy of 3.2 eV, which limits its application from using solar energy [4,15]. In addition, filtration and separation processes are required at the end of the treatment, which increases the costs due to the granulometric size of $TiO_2$ (74 microns) [17].

Some strategies have been proposed in order to overcome these drawbacks. These include the use of supports such as silica [18], zeolites [19], and activated carbon. For

example, the use of activated carbon (AC) as support achieves minimal losses of $TiO_2$. Studies show that AC improves the efficiency of the photocatalytic process thanks to its high adsorption capacity given by its porous structure. AC is a good support, due to its granulometry, hardness, and high surface area [17].

AC develops a synergistic adsorption-degradation effect according to its surface chemistry [17]. AC can oxidize cyanide through the adsorption of molecular oxygen on the AC surface. Adsorbed oxygen reacts with functional groups characteristic of the AC surface to form hydrogen peroxide ($H_2O_2$) that finally reacts with cyanide ion ($CN^-$) to form cyanate, according to Equation (6) [20]. The cyanide oxidation process with AC reached oxidation percentages between 60 and 70% after 8 h [21]. Although AC could adsorb cyanide, preliminary tests have shown that the adsorption percentage of cyanide is less than 5% [22].

$$CN^-_{(aq)} + H_2O_{2(aq)} \rightleftarrows CNO^-_{(aq)} + H_2O_{(aq)} \tag{6}$$

The optimization of the photocatalytic material and the study of the influence of factors such as pH, hydroxyl radicals concentration, or the organic compounds concentration on the cyanide photodegradation have been analyzed to improve the photocatalytic activity [10,14]. Nevertheless, the design of the photocatalytic reactor has been less studied, and there is more information available about the optimization of photocatalytic catalyst. For this reason, new alternatives referring to the photo-reactors configuration are necessary to improve the photocatalytic processes [23].

Batch photo-reactors require long times to achieve significant cyanide removal percentages (>90%); for this reason, the design of photocatalytic reactors in a different configuration than batch is essential. In a batch reactor, the UV light has contact mainly at the surface level, without considerable entry and dissipation within the fluid. The thickness of liquid formed from the base of the reactors decreases the activation of the photocatalyst and consequently the degradation of cyanide [17,24,25].

Another disadvantage of a batch reactor is the difficult separation of catalyst after the degradation process. This could be tackled by the implementation of a continuous reactor in the treatment of pollutants [26]. In a previous study, a multi-phase continuous flow reactor was tested in the photocatalytic oxidation of cyanide using $TiO_2$ as a photocatalyst. Then, the reactor was scaled up to degrade cyanide on an industrial level [27].

This investigation is oriented to the use of composites of titanium dioxide impregnated on active carbon ($TiO_2$-AC) as photocatalyst for the degradation of cyanide in a continuous flow photo-reactor. The use of the $TiO_2$-AC composite and the design of the continuous photo-reactor could enhance the photocatalytic oxidation performance. In addition, this strategic photo-reactor configuration can replace the conventional agitated batch system and reaches high cyanide degradation percentages.

## 2. Results

### 2.1. Physical and Chemical Characterization of GCR-20 Activated Carbon and $TiO_2$-AC Composite

The granular composite used as photocatalyst was obtained by the wet impregnation of $TiO_2$ over AC. A porous network was observed in SEM micrographs of AC and $TiO_2$-AC composite (Figure S1, Supplementary Materials). The content of $TiO_2$ on the AC was analyzed by ICP-OES, and the impregnation was 0.27% *w/w*.

The textural properties determined by $N_2$ physisorption and BET (Brunauer-Emmett_Teller) modeling listed in Table 1 show that AC support and $TiO_2$-AC catalyst had more than 900 $m^2$/g of specific surface area. It indicates that there exists available AC porosity for the adsorption of cyanide ion. ASTM (American Society for Testing and Materials) analysis results for $d_{80}$ particle size, humidity, volatile, ashes, and fixed carbon listed in Table 2 show that the support material is resistant to thermal and mechanical environments, which are conditions that make the AC a good support and synergic effect material to oxidize cyanide, since it enables work with a clear fluid process. Meanwhile, the granular catalyst (3.10 mm) was immobilized avoiding following recovery operations [28].

**Table 1.** Textural properties determined by $N_2$ physisorption and BET modeling of AC and $TiO_2$-AC catalyst.

| Material | $S_{BET}$ (m²/g) | Pore Volume (cm³/g) | Φ (Å) |
|---|---|---|---|
| AC | 1336 | 0.618 | 58.92 |
| $TiO_2$-AC | 902 | 0.504 | 33.97 |

**Table 2.** Physical and chemical properties of AC.

| Parameter | Value |
|---|---|
| Particle size $d_{80}$ (mm) | 3.10 |
| Humidity (%) | 6.82 |
| Volatile (%) | 5.79 |
| Ashes (%) | 7.85 |
| Fixed Carbon (%) | 79.55 |

### 2.2. Photo-Reactor Construction

The investigation was performed in a continuous flow glass reactor with irradiation of UV lamps. $TiO_2$-AC catalysts were added into the reactor using nylon material nets to support the composite in several beds. We selected a reactor design with the maximum proximity between UV lamps and a stable and consistent flow into the reactor. The configuration of the used material is summarized in Scheme 1.

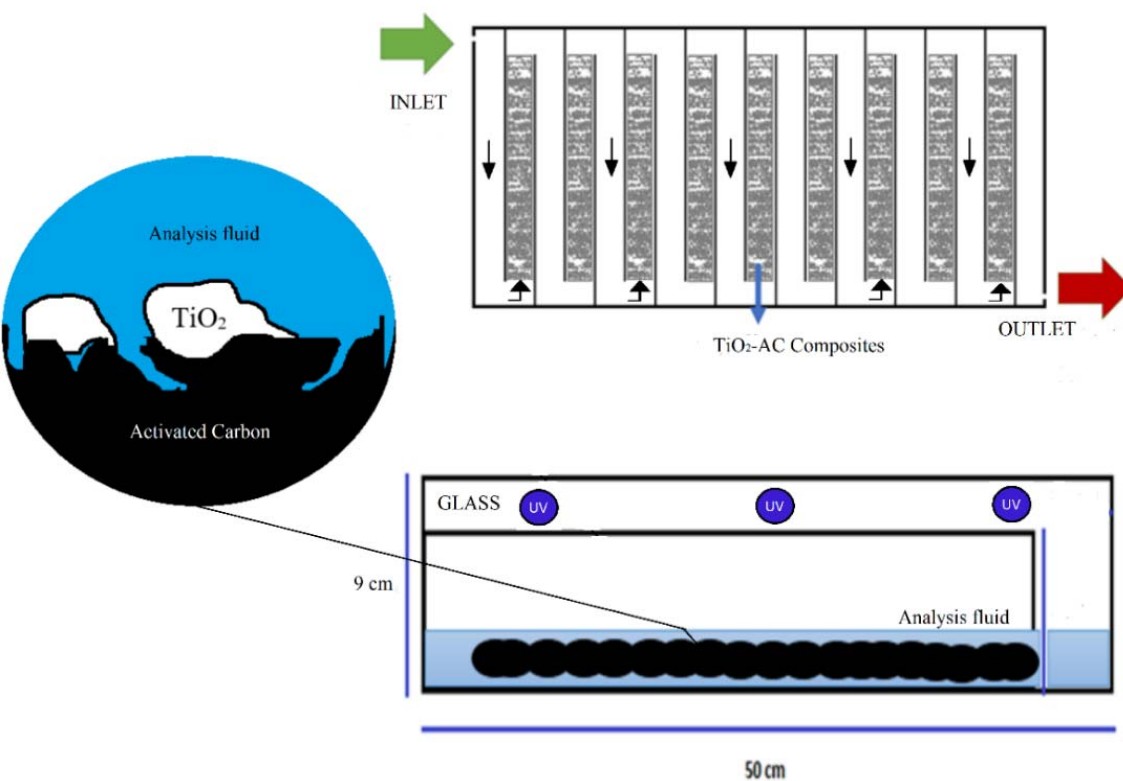

**Scheme 1.** Distribution of immobilized beds of AC or $TiO_2$-AC during the adsorption process and photocatalytic degradation of the cyanide ion in the continuous flow photo-reactor.

Then, the main configuration conditions were a continuous flow of 6.60 mL/s of cyanide solution generated by a 120 rpm peristaltic geopump, 8 mm height layer liquid, nine immobilized beds of AC or $TiO_2$-AC, and a total recirculation volume of 5 L.

Since the adsorption and photocatalytic degradation of cyanide can occur simultaneously, two stage-tests were performed in order to study adsorption and photocatalytic degradation individually. In addition, a simultaneous process was carried out.

### 2.3. Cyanide Adsorption Tests

Through tests of adsorption of $CN^-$ under dark conditions (no UV irradiation) on the AC and $TiO_2$-AC composites, a required adsorption equilibrium time of 24 h was determined.

We found that the cyanide adsorption study had a better fit to the linearized mathematical model of the Langmuir isotherm for both AC and $TiO_2$-AC (results summarized in Table 3). The results indicated that the $TiO_2$-AC maximum cyanide adsorption capacity was lower (1/3) than AC. This finding agreed with the textural properties, since the specific surface area of AC decreased once $TiO_2$ was impregnated. Nonetheless, the adsorption and the energy associated to the process had similar behavior, as shown in Figure 1.

**Table 3.** Parameters calculated for Langmuir isotherm model.

| Parameter | AC | $TiO_2$-AC |
|---|---|---|
| $q_{max}$ (mg·g$^{-1}$) | 155.17 | 52.33 |
| b (L·mg$^{-1}$) | 0.013 | 0.015 |
| $R^2$ | 0.99 | 0.95 |

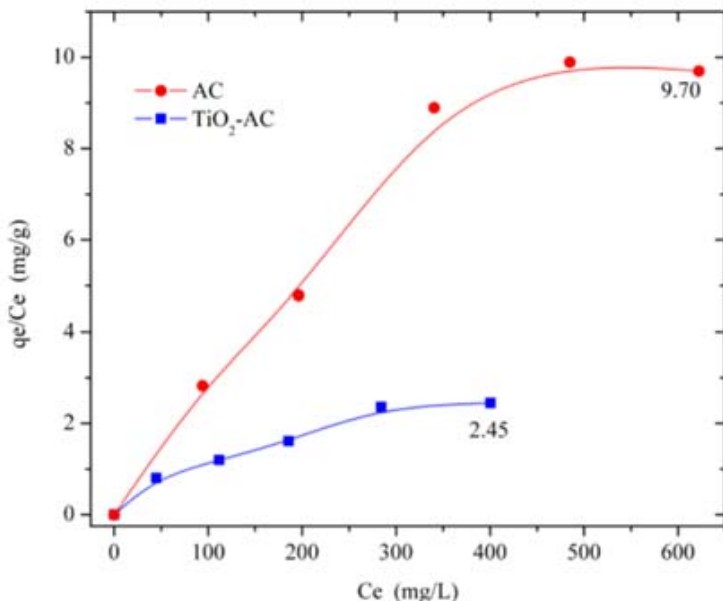

**Figure 1.** Langmuir adsorption isotherms of cyanide (T = 20 °C; 1 g $TiO_2$-AC/L).

Individually, the adsorption of cyanide ion tests were performed for three concentrations 30 g/L, 45 g/L, and 60 g/L of $TiO_2$-AC in 500 mg/L synthetic NaCN solutions, under dark conditions in the continuous flow reactor during 24 h at a pH of 10.5 and 20 °C. Another assay was carried out with 60 g/L of AC at the same conditions. The results listed in Table 4 demonstrate that AC (60 g/L) reached 78% of cyanide degradation due to the adsorption process. It is assumed that adsorption and oxidation processes simultaneously occurred due to oxidant species formed during the process with the dissolved oxygen and functional groups of the AC surface.

**Table 4.** Kinetic modeling of cyanide ion adsorption (pseudo-second order).

| Parameter | AC 60 g/L | TiO$_2$-AC 30 g/L | TiO$_2$-AC 45 g/L | TiO$_2$-AC 60 g/L |
|---|---|---|---|---|
| $K_{app}$ (g·g$^{-1}$min$^{-1}$) | $5.42 \times 10^{-6}$ | $1.66 \times 10^{-6}$ | $3.98 \times 10^{-6}$ | $4.43 \times 10^{-6}$ |
| $R^2$ | 0.98 | 0.83 | 0.98 | 0.98 |
| CN$^-$ Degradation due to adsorption (%) | 78.06 | 57.41 | 71.34 | 74.95 |

On the other hand, TiO$_2$-AC catalyst in 60 g/L concentration reached 74% of CN$^-$ degradation. The cyanide adsorption in the performed tests show similar trends (Figure 2a) and kinetics show that adsorption fits as a pseudo-second-order model through higher correlation coefficients calculation (Table 4). These results indicate that a chemisorption is possible, as suggested by Eskandari [29].

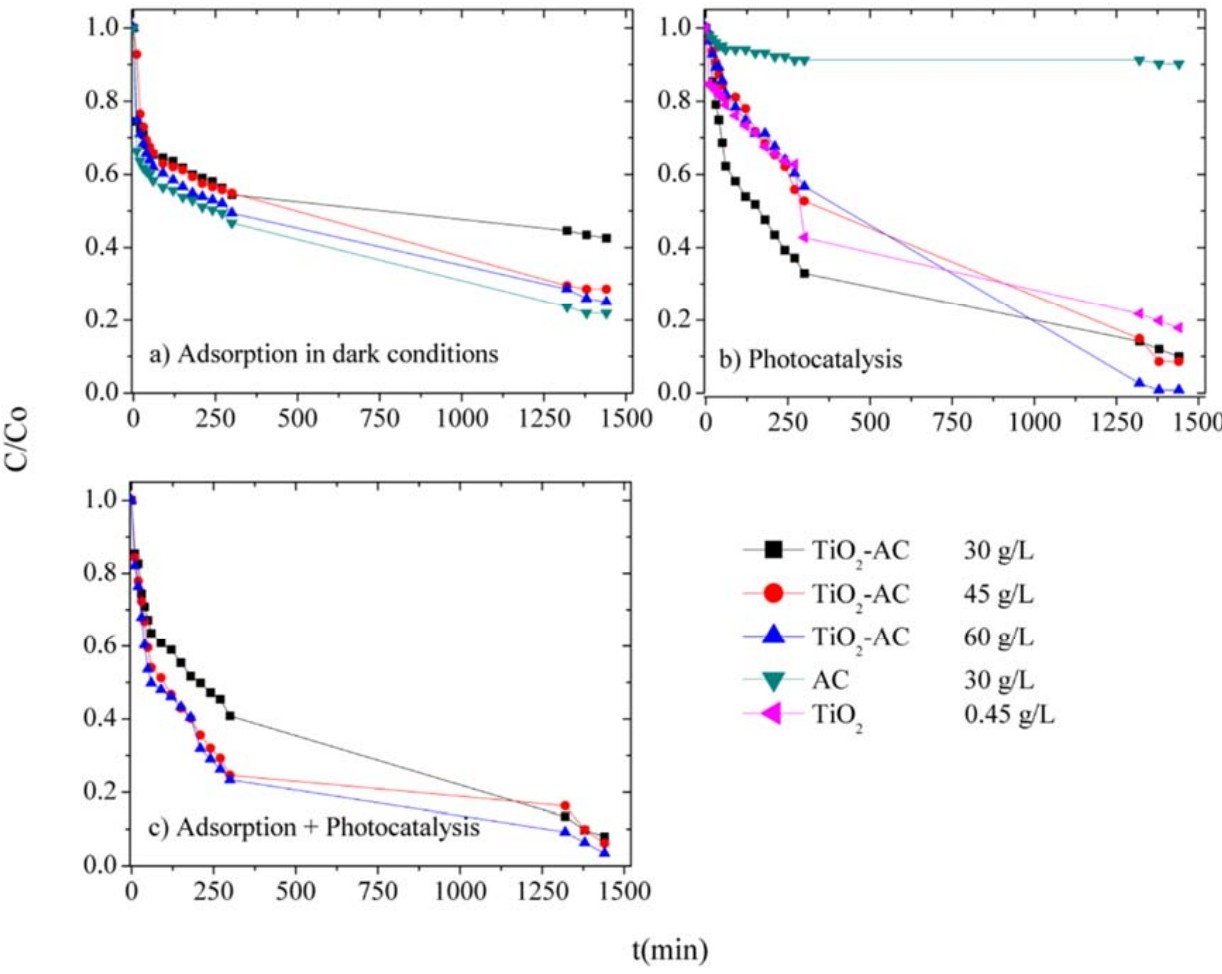

**Figure 2.** Cyanide ion degradation by (**a**) adsorption, (**b**) photocatalysis, and (**c**) simultaneous adsorption and photocatalysis.

### 2.4. Photocatalytic Cyanide Ion Degradation

Once the cyanide ion adsorption was performed during 24 h under dark conditions, the photocatalytic tests started by turning on the UV lamps. Tests were carried out at pH 10.5 with three concentrations 30, 45, and 60 g/L (tested in adsorption) during 24 h. More than 90% of cyanide ion was degraded in all assays using TiO$_2$-AC composites (Table 5). In order to compare the photocatalytic activity performance of materials used for composites on CN$^-$ degradation, tests with TiO$_2$ and AC were performed separately. Although we focused on the TiO$_2$-AC photocatalytic performance, AC yielded an unexpected 9.73% of cyanide ion degradation. Based on other investigations of AOPs with hydroxyl radicals,

this result was explained by photocatalysis promoted by oxygen peroxide. This mechanism considers that $H_2O_2$ can be formed in an unstable and very fast chemical interaction of $O_2$, $H^+$, and $e^-$ in an aqueous medium [29,30]. Moreover, the kinetics obtained for the system was pseudo-first order for all tests performed. $TiO_2$-AC 60 g/L composite assay aimed at 99.16% of cyanide ion degradation; meanwhile, 82.11% was obtained using $TiO_2$ 0.45 g/L (equivalent mass of $TiO_2$ in the $TiO_2$-AC 60 g/L composite).

**Table 5.** Kinetic model results in cyanide ion photodegradation.

| Parameter | AC 60 g/L | TiO₂ 0.45 g/L | TiO₂-AC 30 g/L | TiO₂-AC 45 g/L | TiO₂-AC 60 g/L |
|---|---|---|---|---|---|
| Individual photocatalytic degradation | | | | | |
| $K_{app}$ (min$^{-1}$) | $4.02 \times 10^{-5}$ | $1.06 \times 10^{-3}$ | $1.34 \times 10^{-3}$ | $1.60 \times 10^{-3}$ | $4.90 \times 10^{-2}$ |
| $R^2$ | 0.97 | 0.97 | 0.91 | 0.99 | 0.98 |
| $CN^-$ Degradation (%) | 9.73 | 82.11 | 90.08 | 91.39 | 99.16 |
| Simultaneous adsorption and photocatalytic degradation | | | | | |
| $K_{app}$ (min$^{-1}$) | - | - | $1.47 \times 10^{-3}$ | $1.39 \times 10^{-3}$ | $1.75 \times 10^{-3}$ |
| $R^2$ | - | - | 0.83 | 0.98 | 0.98 |
| $CN^-$ Degradation (%) | - | - | 92.04 | 93.86 | 96.60 |

For comparison reasons, both processes adsorption and photocatalytic cyanide ion degradation were performed in simultaneous assays with the same $TiO_2$-AC concentrations established in this study. As shown in Figure 2, more than 90% of CN- was removed from the solution. In 24 h, 60 g/L of $TiO_2$-AC removed 97% of CN-under UV lights radiation in the continuous flow reactor. If this value is compared to the process that consisted of 24 h of adsorption (75%) and 24 h of photocatalysis (99%), a synergic effect between materials and phenomena appeared. The kinetics for simultaneous tests were pseudo-first order as well as an individual photocatalytic process, as shown in Table 5. Conversely, the kinetics of the adsorption process was not similar as simultaneous study kinetics.

## 3. Discussion

In order to test the adsorption and photocatalysis oxidation of cyanide ion in a continuous flow, a photo-reactor was fabricated based on previous investigations [19,30,31]. Scheme 1 summarizes the design of the photoreactor for $CN^-$ degradation, which contributed to enhance the catalyst-fluid contact. Thus, we used a stable continuous flow of 6.60 mL/s of NaCN synthetic solution with an 8 mm liquid layer, inert material, and a maximum proximity between the UV source and material-fluid. These conditions indicated the advantage of AC and $TiO_2$ performance as a composite, because light can penetrate better in the composite. Since the photoreactor design considered a hollow box with a flow generated by a peristaltic pump through nine separated composite beds, $O_2$ from air can be introduced during all the processes, contributing to enhance the adsorption and photocatalytic oxidation of $CN^-$. Moreover, UV lamps radiation interacted directly with the fluid and the composite. On the other hand, granular AC (derived of coconut shell by steam activation) showed a high specific surface of 1336 m$^2$/g, which would favor the adsorption of contaminants such as $CN^-$. Indeed, the resistant properties of AC to thermal and mechanical processes and a 3.1 mm grain diameter allow building immobilized beds of the composite inside the photoreactor, avoiding post-recovery difficulties. The diameter of AC was smaller than $TiO_2$ grain size (170 nm), and SEM-EDX results showed that $TiO_2$ occupied the external surface of AC, allowing a major contact between UV light source and $TiO_2$ on the support [32]. The specific surface area of the $TiO_2$-AC composite decreased to 902 m$^2$/g due to the $TiO_2$ impregnation However, this result represents a high specific surface area available for the adsorption of cyanide ions. Based on the literature,

the semiconductor would occupy the meso and macro porosity of AC; meanwhile, the microporosity is not affected during the wet impregnation of titania [33,34].

Adsorption and photocatalytic oxidation occurred simultaneously during the treatment of cyanide solutions. Thus, we performed two-stage tests to evaluate each process separately. Both processes were studied at 20 °C and pH = 10.5. During adsorption batch tests, 24 h were obtained as an equilibrium time (when adsorption does not continue). In addition, the Langmuir isotherm model was the better fit to the adsorption for AC and TiO$_2$-AC, where q$_{max}$ was reduced from 155 to 52 mg CN$^-$/g TiO$_2$-AC once the impregnation technique was performed. The CN$^-$ adsorption in AC was drastically reduced once titania was supported, because the TiO$_2$ added to the external surface of AC occupies large holes in the support. In the continuous flow reactor tests, we could estimate that the CN$^-$ adsorption kinetic follows a pseudo-second-order reaction. This result showed that a chemisorption would take place as well as a physisorption of CN$^-$ in AC and TiO$_2$-AC.

Results over 70% of CN$^-$ oxidation in AC and composites revealed that in the process, in light absence, not only adsorption takes place but also an oxidation of cyanide ion could be carried out due to the –OH bonds of the AC surface and existing O$_2$ in aqueous pumping media [29,35].

As expected, granular TiO$_2$-AC composites showed a higher degradation of cyanide ion according to the amount introduced to the continuous flow system. Kinetics and CN$^-$ degradation indicated the following order: TiO$_2$-AC 30 g/L < TiO$_2$-AC 45 g/L < TiO$_2$-AC 60 g/L < AC 60 g/L. Since pH 10.5 was performed in all tests, a negative charge of AC surface was expected, giving a repel effect with cyanide ion. Nonetheless, 75% and 78% of CN$^-$ degradation were determined in light absence using TiO$_2$-AC 60 g/L and AC 60 g/L, respectively.

In photocatalytic tests, as an individual process study, more than 90% of CN$^-$ removal was determined using TiO$_2$-AC composites. Thus, external TiO$_2$ on the AC surface contributed to degrade pollutants in a continuous flow system with composites immobilized with UV 15 W lamps irradiation. For comparison reasons, TiO$_2$ and AC were tested separately in weight amounts corresponding to 60 g/L TiO$_2$-AC. The results listed in Table 5 showed that titania aimed at 82% of CN$^-$ removal; meanwhile, a surprising result of 10% CN$^-$ degradation was obtained with AC. Although AC was not considered as a photo-catalyst, it showed CN$^-$ removal due to the operational conditions and continuous flow system, and H$_2$O$_2$ could be formed rapidly during UV irradiation once the pair e$^-$/h$^+$ is formed [32,36]. Some investigations indicate that hydrogen peroxide can contribute to pollutant degradation in AOPs, since this compound can be formed in an electrochemical process associated to UV incidence and dissolved oxygen presence during the pumping process. These results ensure that TiO$_2$-AC composites enhanced the photocatalysis, increasing the synergic effect between AC and TiO$_2$. Kinetics and cyanide degradation were determined in order 60 g/L AC < 0.45 g/L TiO$_2$ < TiO$_2$-AC 30 g/L < TiO$_2$-AC 45 g/L < TiO$_2$-AC 60 g/L.

Simultaneous adsorption and photocatalysis tests were performed on UV radiation. More than 90% of CN$^-$ was removed in all tests. However, in a comparison analysis of each concentration composite dosage, a numerical variation was detected. For 60 g/L of TiO$_2$-AC, 96% of CN$^-$ removal was obtained during 24 h, where adsorption and photocatalysis took place at the same time. Whereas 74% and 99% of CN$^-$ degradation were obtained in 24 h adsorption and 24 h of photocatalysis, respectively. Thus, a simultaneous process is suitable for cyanide ion degradation under operational continuous flow photorector design. Both photocatalysis and the simultaneous process (photocatalysis + adsorption) correspond to a pseudo-first-order reaction. This result is in concordance with the literature, which attributed the oxidation of pollutants by hydroxyl radical activity. The success of CN$^-$ degradation was enhanced by TiO$_2$-AC presence and the design parameters of the photoreactor, because a stable system could be formed under a near interaction between the UV source and materials of this study.

## 4. Materials and Methods

### 4.1. TiO$_2$-AC Composites Preparation

TiO$_2$-AC composites were prepared by the wet impregnation of TiO$_2$ on activated carbon (AC). First, the AC was washed with distilled water under magnetic stirring, until a clarified washing solution was obtained. Then, 1 g of commercial TiO$_2$ (United States Pharmacopeia (USP) reference standard, 99% anatase) and 100 g of commercial activated carbon (CALGON GRC-20®) were added to 350 mL of distilled water under vigorous stirring for 2 h. The solvent was removed by evaporation, and composites were washed with distilled water several times. Finally, TiO$_2$-AC composites were dried at 110 °C for 2 h [37,38].

### 4.2. Physical and Chemical Characterization of GCR-20 Activated Carbon and TiO$_2$-AC Composite

Standardized sieves were used to determine the particle size distribution of AC by the Standard Test Method for Particle Size Distribution of Granular Activated Carbon ASTM-D2862. Moisture, volatile material, ash, and fixed carbon content were measured for the AC support according to ASTM-D3173 (Standard Test Method for Moisture in the Analysis Sample of Coal and Coke), ASTM-D3175 (Standard Test Method for Volatile Matter in the Analysis Sample of Coal and Coke), and ASTM-D3174 (Standard Test Method for Ash in the Analysis Sample of Coal and Coke from Coal).

The textural properties of the GCR-20 AC and TiO$_2$-AC composites were determined by N$_2$ physisorption isotherms in a Quantachrome Instruments Nova 4200e (Quantachrome Instruments, Boynton Beach, FL, USA). The BET model was used to determine the specific surface area.

A scanning electron microscopy analysis (SEM) was performed to analyze the TiO$_2$-AC composite texture with a Vega TESCAN (TESCAN, Brno, Czech Republic) microscope equipped with secondary electron (SE) and backscattered electron (BSE) detectors. The chemical mapping was determined by an energy-dispersive analysis of the X-ray (EDX) using Vega TESCAN scanning electron microscopy attached with a Bruker XFlash 5010 Detector (Bruker, Billerica, MA, USA), with an accelerating voltage of 20 kV under vacuum.

In order to determine the impregnation of TiO$_2$ on the activated carbon, all composites underwent microwave acid digestion with HNO$_3$, HF, and HCl. Titanium content in the acid solution was analyzed by inductively coupled plasma optical emission spectrometry (ICP-OES) (Perkin Elmer Optima 8000, Perkin Elmer Inc., Waltham, MA, USA).

### 4.3. Photo-Reactor Construction

A continuous photo-reactor was made of glass and covered with a wood box to isolate the fluid of the external conditions. Three 15 W UV lamps of 43.74 cm were attached to the lid of the reactor in parallel distribution in order to penetrate as far as possible the fluid and obtain the greatest proximity between the UV lamps, the photo-catalyst, and the fluid. The internal part of the photo-reactor made of glass is a rectangle of 135 cm × 50 cm with walls of 9 cm. Inside of this structure glass, plates of 46 cm × 4 cm were placed perpendicularly, as shown in Scheme 1. A 5-degree angle was adapted between the reactor and the horizontal level in order to maintain stable circulation of the fluid through the path formed inside the reactor.

Constant flow rate in the reactor, thickness of the liquid, residence time of the cyanide solution, and concentration of the catalyst/composite were measured to determine the optimal operating conditions.

### 4.4. Cyanide Adsorption Study

Sodium cyanide solutions with concentrations of 200, 400, 600, 800, and 1000 mg/L were prepared. The adsorption tests were carried out in a batch with 50 mL of sodium cyanide solution and 0.05 g of each AC and TiO$_2$-AC composite under continuous stirring for 24 h. The data of cyanide ion adsorption were studied using the mathematical model of the Langmuir isotherm. After adsorption-desorption equilibrium was achieved, the solid

was filtered, and the cyanide ion concentration (in solution) was determined by titration with a 0.2256 N AgNO$_3$ solution [28].

The data were studied and modeled by the Langmuir isotherm according to the mathematical relationship described in Equation (7).

$$\frac{C_e}{q_e} = \frac{1}{q_{max}} \times C_e + \frac{1}{q_{max} \times b} \tag{7}$$

where $C_e$ is the cyanide ion equilibrium concentration in the solution (mg/L), $q_e$ is the equilibrium concentration of cyanide ion over the adsorbents (AC or TiO$_2$-AC) (mg/g), $q_{max}$ is the maximum mass of cyanide ion adsorbed per 1 g of adsorbent (AC or TiO$_2$-AC) (mg/g), and b is the independent variable referring to the free energy of adsorption (L/mg).

### 4.5. Adsorption Study

### 4.5.1. Adsorption with AC

Adsorption tests were carried out in a photo-reactor of continuous flow under dark conditions at room temperature. A sodium cyanide solution of 500 mg/L was recirculated through a peristaltic pump (Geopump Inc., Medina, NY, USA) for 24 h. The pH of solution was adjusted at 10.5 throughout the adsorption experiments with the addition of NaOH. Concentrations of 30, 45, and 60 g/L of activated carbon were used for each test. Samples of 5 mL were taken each 10 min during the first hour and then each 30 min for three hours. The solution was recirculated overnight, and the next day, samples of 5 mL were taken each hour for 2 h. The cyanide ion concentration was determined by titration with a 0.2256 N AgNO$_3$ solution.

### 4.5.2. Adsorption with TiO$_2$-AC Composite and Photodegradation Process

Cyanide ion adsorption on AC and the photodegradation of cyanide ion occur simultaneously. Experiments of cyanide ion adsorption on the TiO$_2$-AC composite were carried out under the same conditions as in the adsorption with AC. At the end of the adsorption experiments, the photodegradation of cyanide ion was evaluated for 24 h under UV light. Samples of 5 mL were taken (in the time intervals explained in Section 4.5.1) to determine the cyanide ion concentration by titration with a 0.2256 N AgNO$_3$ solution.

### 4.6. Photocatalytic Cyanide Ion Degradation

Photocatalytic activity was performed in a continuous photo-reactor using an initial concentration of sodium cyanide of 500 mg/L and a composite concentration of 30, 45, and 60 g/L at pH 10.5 and room temperature. The system (Scheme 1) was maintained in dark conditions during 24 h to ensure adsorption-desorption equilibrium. Then, UV lamps (Philips, Amsterdam, The Netherlands, T8 G13 UV-C 15 W) were turned on to initiate the photodegradation. Aliquots of 5 mL were taken (in the time intervals explained in Section 4.5.1) to analyze the residual cyanide ion by titration with a 0.2256 N AgNO$_3$ solution.

Cyanide ion photodegradation with AC (saturated with cyanide) and TiO$_2$ of 60 g/L and 0.45 g/L (corresponding to the impregnation percentage) respectively were carried out in order to compare with the photodegradation results with the composite.

After the photodegradation test, samples of the remaining solution were taken to determine the concentration of titanium and sodium dissolved by atomic absorption spectroscopy.

## 5. Conclusions

Within this study, we built and performed a continuous flow photo-reactor during cyanide ion degradation using TiO$_2$-AC composites. Design operational conditions were established into the photoreactor where a minimum distance between UV source and fluid/composites and a stable continuous flow though immobilized composites were achieved. Therefore, inside of the reactor, adsorption and photocatalysis of CN$^-$ synthetic solutions were performed using TiO$_2$-AC in concentrations of 30, 45, and 60 g/L. When

both processes, adsorption and photocatalysis, were studied separately, the adsorption of CN$^-$ on AC in light absence decreased when TiO$_2$-AC was performed, achieving 75% of CN$^-$ removal. Conversely, TiO$_2$-AC composites markedly enhance the photocatalytic process in comparison to individual TiO$_2$ and AC performances, achieving 99% of cyanide ion degradation. During 24 h of simultaneous process, 96% of CN$^-$ was aimed. In the success operational conditions of the photoreactor, we determined that AC could contribute to photocatalysis cyanide degradation, since AC showed 10% of CN$^-$ removal with UV irradiation. By other hand, in light absence, granular AC tested showed a considerable amount of cyanide degradation since –OH bonds on the AC surface and O$_2$ added to the system could contribute to cyanide removal. Thus, this reactor design and TiO$_2$-AC would represent an encouraging alternative of cyanide degradation in a continuous flow system due to their synergic effect in wastewaters remediation.

**Supplementary Materials:** The following are available online at https://www.mdpi.com/article/10.3390/catal11080924/s1, Figure S1. (a) SEM micrograph of AC, 783x; (b) SEM micrograph of TiO$_2$-AC composite, 927x; (c) EDX mapping performed on the micrograph of the TiO$_2$-AC composite, 927x, Figure S2. EDX mapping performed on a micrograph of the TiO$_2$-AC composite, 783x.

**Author Contributions:** Conceptualization, S.C. and D.E.; methodology, S.C., A.B.L. and D.E.; formal analysis, S.C., A.B.L.; investigation, S.C., A.B.L., E.d.l.T. and D.E.; resources, E.d.l.T. and D.E.; data curation, S.C., A.B.L.; writing—original draft preparation, S.C., A.B.L., L.E.M.-P.; writing—review and editing, S.C., A.B.L., L.E.M.-P. and D.E.; visualization, S.C., A.B.L., L.E.M.-P., E.d.l.T. and D.E.; supervision, L.E.M.-P., E.d.l.T. and D.E.; project administration, E.d.l.T. and D.E. All authors have read and agreed to the published version of the manuscript.

**Funding:** This research received no external funding.

**Acknowledgments:** The authors would like to thank the financial support provided by Escuela Politécnica Nacional through the project PII-DEMEX-20-01 belonging to the Master of Research in Metallurgy.

**Conflicts of Interest:** The authors declare no conflict of interest.

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
