# Peer review of "Photocatalytic Study of Cyanide Oxidation Using Titanium Dioxide (TiO2)-Activated Carbon Composites in a Continuous Flow Photo-Reactor"

_catalysts, doi:10.3390/catal11080924_

Round 1

Reviewer 1 Report

Most Respected Authors,

I have read carefully the present manuscript and I found it to be interesting to publish within this journal. However, the investigation is not yet complete. In my view it needs two more intermediate samples which have different kind of AC concentration within the composite. Moreover, P25 contains at least 10% of rutile (in the manuscript 99% is given), hence XRD analysis is needed for both the pure and for the composite as well. Best regards,

The referee

Reviewer 2 Report

The work in the presented form is not suitable for publication. Grave experimental errors were made in two methods (FTIR and SEM-EDX).

Severe mistakes

  1. Conclusions from the SEM-EDX method do not follow the data presented in Fig. 1. There is a lack of qualitative and quantitative data from EDX. The presented figures do not confirm the conclusions. Please provide correct results and interpret them well.

Line 126-131 „The composite was chemically analyzed by SEM-EDX analysis images (Fig-1), where it is shown that the semiconductor material was deposited on the largest 128 holes on the AC surface. Therefore, TiO2 located on the external surface of AC contributes 129 to the interaction between UV irradiation and the photo-catalyst.”

  1. FTIR spectra are incorrectly made. In the spectra of Fig. 2 and Fig. 4, there are practically no differences for "AC", "TiO2-AC" and "AC Post-adsorption". There is no visible band characteristic for TiO2 and AC in the spectra. There is only one significant 1360 cm-1 band that has not been interpreted. The 3426 cm-1 band probably comes from the atmosphere (OH oscillation in H2O). FTIR results are misinterpreted. Please provide correct results and interpret them well.

Minor mistakes

  1. Please complete the data on publications [28] and [29]. It is not known what kind of publication, doctoral or master's thesis is?
  2. In the description of research methods p. 4.2 in lines 341-342, there is no information about the EDX detector, and in line 127 it is stated, it was analyzed by the SEM-EDX method.
  3. In the description of the research methods p. 4.2 in lines 344-346, there is no information in what form the samples were recorded and what measurement technique was used (ATR, DRIFT, transmission).

Author Response

Thank you very much for your comments. Please see the attachment for the responses.

Best regards.

Reviewer 3 Report

The authors presented in this study a strategy to degrade cyanides by combining both adsorption and photodegradation in a continuous flow reactor. The study appears systematic and comprehensive, offering an interesting engineering perspective on the remediation of cyanides. The addition of a reusability study and a better formatting could really strengthen the clarity and the scientific impact of the current paper, hence a major revision is recommended on the basis of the following comments:

  • The manuscript should be carefully proofread as there are some typos in all sections. See for instance line 397: 30, 45 y 60 397 g/L at pH 10.5
  • The authors have provided a significant context to their study by presenting some real case-studies, such as the gold mining industry in Ecuador. Are cyanides the only challenge for this type of wastewater? The complexation of metals with cyanides seem to represent an additional challenge that the authors could at least mention in their introduction. See DOI: 10.1016/j.jhazmat.2021.125802
  • The formatting of some figures should be improved. In figure 1, for instance, the scale bars should be clearly indicated. What is figure 1.c showing?
  • It is suggested that figure 4a, 5 and 6 be combined in a single, composite figure to better visualize the different processes and the synergism between photocatalysis and adsorption.
  • In figure 5, why is TiO2-AC-30 (black dots) faster than all the other catalysts over the first 4 hours?
  • Can the authors elaborate further on the expected degradation mechanism/pathway under adsorption/photocatalysis conditions?
  • Please clarify the total bed volume and the frequency of aliquot withdrawal.
  • It is recommended that the authors conduct a reusability study. Does the performance of the catalyst degrade over multiple tests? This is a critical performance parameter in photocatalysis and TiO2 leaching might occur over time.

Author Response

(The authors gave the same response as above.)

Round 2

Reviewer 1 Report

Dear Authors,

The paper can be now published

Author Response

No comments. Thank you very much.

Reviewer 2 Report

The authors introduced minor corrections which improved the quality of the publication. However, major bugs in the SEM-EDX and FTIR results have not been fixed. It is not about introducing corrections to the presented results but about a complete change of the measurement methodology and the presented results.

The presented results concerning these methods are unacceptable and disqualify the manuscript for publication.

The first comments on minor mistakes have not been corrected.

1. Conclusions from the SEM-EDX method do not follow the data presented in Fig. 1. There is a lack of qualitative and quantitative data from EDX. The presented figures do not confirm the conclusions. Please provide correct results and interpret them well.

Line 126-131 „The composite was chemically analyzed by SEM-EDX analysis images (Fig-1), where it is shown that the semiconductor material was deposited on the largest 128 holes on the AC surface. Therefore, TiO2 located on the external surface of AC contributes 129 to the interaction between UV irradiation and the photo-catalyst.”

2. FTIR spectra are incorrectly made. In the spectra of Fig. 2 and Fig. 4, there are practically no differences for "AC", "TiO2-AC" and "AC Post-adsorption". There is no visible band characteristic for TiO2 and AC in the spectra. There is only one significant 1360 cm-1 band that has not been interpreted. The 3426 cm-1 band probably comes from the atmosphere (OH oscillation in H2O). FTIR results are misinterpreted. Please provide correct results and interpret them well.

Reviewer 3 Report

I am happy with the amendments introduced by the authors, in particular regarding the methods.

Author Response

No comments. Thank you very much.

Round 3

Reviewer 2 Report

Figures 1 and 2 do not add any information to the article. From my first comment, I point out that these results cannot be included in the article. They need to be changed or deleted. The authors improved the discussion a bit, but these corrections did not produce positive changes. The article is unpublishable in its present form as Figures 1 and 2 will be misleading to readers. This is a typical example of how data (spectra IR and SEM-EDX results) should not be presented.

Author Response

Thank you very much for your suggestions.
